# Replicate Testing of Clinical Endpoints Can Prevent No-Go Decisions for Beneficial Vaccines

**DOI:** 10.3390/vaccines11091501

**Published:** 2023-09-19

**Authors:** Daniel I. S. Rosenbloom, Julie Dudášová, Casey Davis, Radha A. Railkar, Nitin Mehrotra, Jeffrey R. Sachs

**Affiliations:** 1Quantitative Pharmacology and Pharmacometrics, Merck & Co., Inc., Rahway, NJ 07065, USA; daniel.rosenbloom@merck.com (D.I.S.R.); casey.davis14@gilead.com (C.D.); nmehrotra@alnylam.com (N.M.); 2Quantitative Pharmacology and Pharmacometrics, MSD Czech Republic, 15000 Prague, Czech Republic; 3First Faculty of Medicine, Charles University, 12108 Prague, Czech Republic; 4Biostatistics and Research Decision Sciences, Merck & Co., Inc., Rahway, NJ 07065, USA

**Keywords:** clinical trial design, vaccine efficacy, diagnostic assays, case-counting, false-positive rate, diagnostic error, test error, mathematical modeling, misclassification bias

## Abstract

In vaccine efficacy trials, inaccurate counting of infection cases leads to systematic under-estimation—or “dilution”—of vaccine efficacy. In particular, if a sufficient fraction of observed cases are false positives, apparent efficacy will be greatly reduced, leading to unwarranted no-go decisions in vaccine development. Here, we propose a range of replicate testing strategies to address this problem, considering the additional challenge of uncertainty in both infection incidence and diagnostic assay specificity/sensitivity. A strategy that counts an infection case only if a majority of replicate assays return a positive result can substantially reduce efficacy dilution for assays with non-systematic (i.e., “random”) errors. We also find that a cost-effective variant of this strategy, using confirmatory assays only if an initial assay is positive, yields a comparable benefit. In clinical trials, where frequent longitudinal samples are needed to detect short-lived infections, this “confirmatory majority rule” strategy can prevent the accumulation of false positives from magnifying efficacy dilution. When widespread public health screening is used for viruses, such as SARS-CoV-2, that have non-differentiating features or may be asymptomatic, these strategies can also serve to reduce unneeded isolations caused by false positives.

## 1. Introduction

Randomized clinical trials of vaccine efficacy are designed to estimate the relative reduction in the risk of infection of vaccinated subjects relative to that of control subjects [1]. These trials, which may enroll over 30,000 participants to measure differences in infection incidence between study arms, depend critically on the methods used to count infection cases. If clinical case definitions and diagnostic tests are not accurate, then vaccine efficacy estimates can be misleading. In particular, the inaccurate counting of cases systematically and predictably biases the estimated efficacy toward the null value—either toward zero, in the case of comparison to placebo arm, or toward no differentiation from an active comparator arm [2,3]. This effect is referred to, here, as “dilution” of vaccine efficacy. There is one aspect of dilution that can be mitigated by a pragmatic approach proposed here.

To see why there is dilution, consider, for illustration purposes, a trial with an incidence rate of 2% for the period of the trial, as well as 2000 vaccinated and 2000 placebo participants. If the vaccine has 100% efficacy, then 40 cases will be expected in the placebo arm versus none expected in the vaccine arm. If the diagnostic assay has a 0.01 (1%) false-positive rate (assumed to be equal across study arms), the placebo arm will have an expected 40 true cases (2% of 2000). However, roughly 60 cases are expected to accrue in the placebo arm $\left(40+0.01\times (2000-40)\right)$, and 20 cases (0.01 × 2000) are expected in the vaccine arm (all due to false positives because, in this example, we have assumed the vaccine to be 100% efficacious), yielding a reported efficacy of only about $1-\frac{20}{60}\approx 67\%$. Such a reduction in apparent efficacy can lead to unwarranted no-go decisions in the vaccine development process or to misleading interpretations of vaccine effectiveness studies [4,5,6]. When testing is repeated over time (e.g., to detect asymptomatic infection), dilution is further complicated by false positives that may occur at each repeated test. While this work focuses on vaccine efficacy trials, dilution also applies to any trial that depends on a binary endpoint that uses an imperfect assay.

Lachenbruch [7] proposed an adjustment of the efficacy estimate by accounting for the values of the false-positive rate and false-negative rate of the diagnostic assay. If the specificity and sensitivity of an assay are well understood, this method can lead to a more accurate point estimate of efficacy. This method, however, also increases variance of the estimate, meaning that larger clinical trials are required to achieve the desired study power.

Here, we present—and show the substantial potential advantages of—a series of pragmatic “majority rule” replicate assay strategies, as alternative methods, to manage the problem of vaccine efficacy dilution. These strategies require neither an accurate estimate of assay false-positive and negative rates nor a larger clinical trial. In the context of vaccine development, the goal of this work is to avoid decisions to halt development due to an apparent lack of efficacy when the true underlying efficacy would have warranted continued development. Of course, the timing of the trial, the sampling in the trial, disease prevalence, incidence rates, manufacturing and storage stability, local health care system attributes, regulatory and commercial competition, and many other considerations can go into these decisions, which can, thus, be nuanced and difficult. Typical efficacy-based decisions in vaccine development programs may require, for example, that the lower bound of a 95% confidence interval for efficacy be above 0 for a proof of concept study or above 25% for a monovalent vaccine in Phase 3. Many considerations can alter these criteria. For example, where multiple vaccines are tested simultaneously, to determine adequate safety and efficacy for rapid and broad uptake, more stringent criteria may be used: a lower bound threshold of 30% has been applied to COVID-19 vaccine studies [8].

After presenting calculations that elucidate the efficacy dilution, resulting from single and replicate assay strategies, we demonstrate that a resource-sparing strategy using repeated assays to confirm an initial positive result achieves greater specificity without a meaningful loss of sensitivity. We then simulate a vaccine efficacy trial to demonstrate how this resource-sparing strategy can resolve trade-offs in the design of clinical trials that require frequent sampling to detect infection cases. The software (an R Shiny app) provided in the supplement can assist investigators in the development of replicate assay strategies to optimize efficacy trials and minimize the chance of unwarranted no-go decisions for clinically beneficial vaccines.

## 2. Materials and Methods

### 2.1. Efficacy Dilution Calculation

We express the impact of false positives and/or negatives as the relative error in efficacy estimation, the efficacy dilution $D=1-\frac{\mathrm{OVE}}{\mathrm{TVE}}$, where OVE is the observed (estimated) vaccine efficacy and TVE is the true efficacy of the vaccine. In the analyses that follow, we assume that cohorts are sufficiently large enough that deviations from expected values of case counts can be neglected. These deviations can be accounted for in clinical trial simulation, such as the scenario discussed in Section 3.5. Efficacy dilution D can be calculated in terms of other estimated (or assumed) quantities:FP: False-positive rate; the probability that a diagnostic assay reports uninfected individuals as being infected (i.e., as “infection cases”);FN: False-negative rate; the probability that a diagnostic assay reports infected individuals as not being infected (i.e., as “non-cases”);TIP: True infected positives; the true incidence of an infection in the unvaccinated (placebo) arm during the time period of the trial;OIP: Observed infected positives; the apparent incidence in the placebo arm, which can differ from the true value TIP for assays with nonzero false-positive and/or false-negative rates. This quantity can be derived from FP, FN, and TIP (Equation (1), below).

Although we are defining cases in terms of “infection”, they are often defined in terms of disease (symptomatic infection); this difference does not impact results if all definitions are used consistently.

If only symptomatic subjects are tested with the assay, FP and FN can be treated as net error rates, whether the source of the error lies with symptom assessment or the assay. In all calculations, both FP and FN are assumed to be less than 50%, i.e., the test procedure will, more likely than not, provide the correct outcome.

It follows from the above definitions that the expected value for the observed incidence in a placebo arm (OIP) is
(1)$$\mathrm{OIP}=\mathrm{TIP}\left(1-\mathrm{FN}\right)+\left(1-\mathrm{TIP}\right)\mathrm{FP}.$$

All rates must be represented on the same time basis: If FP and FN are expressed per each use of the assay, TIP must be converted to a “per assay occasion” basis. In other words, if study participants are tested for infection four times per year (not counting a baseline test establishing negative status upon study entry)—and if a positive result at any one of these quarterly occasions is treated as a case—then an annual incidence rate would be converted to a quarterly incidence rate for all dilution calculations: $\mathrm{TIP}=1-{\left(1-{\mathrm{TIP}}_{annual}\right)}^{\frac{1}{4}}\approx \frac{{\mathrm{TIP}}_{annual}}{4}$ (approximation valid for small values of TIP).

If FP, FN, and TIP are all relatively small (no more than 0.2), the observed placebo incidence is approximately
(2)OIP≈TIP+FP.

If the true efficacy of a vaccine is TVE, the expected value for observed incidence in a vaccine arm (OIV), assuming that false-positive and false-negative rates are equal across both arms, is
(3)$$\mathrm{OIV}=\mathrm{TIP}\left(1-\mathrm{TVE}\right)\left(1-\mathrm{FN}\right)+\left(1-\mathrm{TIP}\left(1-\mathrm{TVE}\right)\right)\mathrm{FP},$$
or, using the same approximation as in (2),
(4)$$\mathrm{OIV}\approx \mathrm{TIP}\left(1-\mathrm{TVE}\right)+\mathrm{FP}.$$

Assuming a sufficiently large trial cohort such that deviations from the expected value of OIP and OIV are negligible, the observed vaccine efficacy (OVE) is
(5)$$\mathrm{OVE}=1-\frac{\mathrm{OIV}}{\mathrm{OIP}}.$$

Deviations from the expected values can be addressed in simulations of specific clinical scenarios (see Section 2.4 and Section 3.5). The fraction by which inaccuracies in the infection assay dilutes observed efficacy is the dilution effect, D:(6)$$D=1-\frac{\mathrm{OVE}}{\mathrm{TVE}},$$
which becomes, after some simplification,
(7)$$D=\frac{\mathrm{FP}}{\mathrm{OIP}}.$$

Using the approximation in (2), the dilution effect is
(8)$$D=\frac{FP}{OIP}\approx \frac{\mathrm{FP}}{\mathrm{TIP}+\mathrm{FP}}.$$

### 2.2. Replicate Assays and “Majority Rule” Strategy

If multiple repeated runs of an assay can be treated as independent (e.g., not having substantial cross-contamination that impacts multiple samples), we can mitigate the impact of efficacy dilution by requiring multiple assay results to be positive in order to count as a case. The assay strategy is, then, determined by the total number of replicate assays performed (*n*) and how many of these must be positive (*m*) to count the case. The effective false-positive and false-negative rates for this strategy are the binomial probabilities:(9)$$\begin{array}{c} \mathrm{FP}\left(n,m\right)=P\left(\geq m\;\mathrm{false}\;\mathrm{positives}\;\mathrm{occur}\;|\;\mathrm{the}\;\mathrm{sample}\;\mathrm{is}\;\mathrm{truly}\;\mathrm{negative}\right)\\ =\sum_{k=m}^{n}\left(\begin{array}{c} n\\ k \end{array}\right){\mathrm{FP}}^{k}{\left(1-\mathrm{FP}\right)}^{n-k}, \end{array}$$
(10)$$\begin{array}{c} \mathrm{FN}\left(n,m\right)=P\left(\geq \left(n-m+1\right)\;\mathrm{false}\;\mathrm{negatives}\;\mathrm{occur}\;|\;\mathrm{the}\;\mathrm{sample}\;\mathrm{is}\;\mathrm{truly}\;\mathrm{positive}\right)\\ =\sum_{k=n-m+1}^{n}\left(\begin{array}{c} n\\ k \end{array}\right){\mathrm{FN}}^{k}{\left(1-\mathrm{FN}\right)}^{n-k}. \end{array}$$

For a majority rule strategy, *m* is equal to $\left\lfloor \frac{n}{2}+1\right\rfloor$ ($\left\lfloor \cdot \right\rfloor$ is the floor function, i.e., the integer part of the number enclosed). For odd *n*, the summation terms for both of these expressions are equivalent under the replacement of FP by FN and vice versa: The number of individual assay false positives required for an overall false positive (m) is equal to the number of individual assay false negatives required for an overall false negative (n−m+1). In this situation, both of these expressions decline with *n* for any positive FP and FN less than 0.5 (see Appendix A).

The following approximations, which avoid summing up multiple terms, hold for the majority rule strategy (with odd or even *n*) if FP and FN are sufficiently small:(11)$$\mathrm{FP}\left(n,\left\lfloor \frac{n}{2}+1\right\rfloor \right)\approx \left(\begin{array}{c} n\\ \left\lfloor \frac{n}{2}+1\right\rfloor \end{array}\right){\mathrm{FP}}^{\left\lfloor \frac{n}{2}+1\right\rfloor },$$
(12)$$\mathrm{FN}\left(n,\left\lfloor \frac{n}{2}+1\right\rfloor \right)\approx \left(\begin{array}{c} n\\ \left\lfloor \frac{n+1}{2}\right\rfloor \end{array}\right){\mathrm{FN}}^{\left\lfloor \frac{n+1}{2}\right\rfloor }.$$

For *n* = 3 and FP or FN up to 0.2, these approximations overestimate the exact values by no more than 15% (relative error). As *n* increases, smaller FP or FN is required to maintain the same accuracy of these approximations (falling roughly as the inverse of *n*). For odd *n*, both of these approximate expressions reduce to the same form:(13)$$\left(\begin{array}{c} n\\ \left(n+1\right)/2 \end{array}\right)x^{\left(\frac{n+1}{2}\right)},$$
where *x* is replaced with either FP or FN, as appropriate. Appendix A shows the adequacy of these approximations.

### 2.3. Confirmatory Assays and “Confirmatory Majority Rule” Strategy

A confirmatory strategy is also defined by requiring that at least *m* out of *n* assays be positive, as well as that one of these positives must be the initial assay performed. We consider the possibility that this initial assay may have different characteristics (false-positive and negative rates ${\mathrm{FP}}_{1}$ and ${\mathrm{FN}}_{1}$) from the subsequent assays (all with identical false-positive and negative rates ${\mathrm{FP}}_{2}$ and ${\mathrm{FN}}_{2}$). The resulting effective rates of this testing strategy are
(14)$$\begin{array}{c} {\mathrm{FP}}_{\mathrm{conf}}\left(n,m\right)=P\left(\left.\begin{array}{c} First\;assay\;and\geq \left(m-1\right)\;of\\ the\;remaining\;assays\;are\;false\;positives \end{array}\right|\;\mathrm{the}\;\mathrm{sample}\;\mathrm{is}\;\mathrm{truly}\;\mathrm{negative}\right)\\ ={\mathrm{FP}}_{1}\sum_{k=m-1}^{n-1}\left(\begin{array}{c} n-1\\ k \end{array}\right){\mathrm{FP}}_{2}^{k}{\left(1-{\mathrm{FP}}_{2}\right)}^{n-1-k}, \end{array}$$
(15)$$\begin{array}{c} {\mathrm{FN}}_{\mathrm{conf}}\left(n,m\right)==P\left(\left.\begin{array}{c} First\;assay\;is\;false\;negative;or\;first\;assays\;is\;positive\;and\\ \geq \left(n-m+1\right)\;of\;the\;remaining\;assays\;are\;false\;negatives \end{array}\right|\;\mathrm{the}\;\mathrm{sample}\;\mathrm{is}\;\mathrm{truly}\;\mathrm{positive}\right)\\ ={\mathrm{FN}}_{1}+\left(1-{\mathrm{FN}}_{1}\right)\sum_{k=n-m+1}^{n-1}\left(\begin{array}{c} n-1\\ k \end{array}\right){\mathrm{FN}}_{2}^{k}{\left(1-{\mathrm{FN}}_{2}\right)}^{n-1-k}. \end{array}$$

As above in Section 2.2, if a majority are required to be positive (“confirmatory majority rule” strategy), these expressions may be approximated:(16)$${\mathrm{FP}}_{\mathrm{conf}}\left(n,\left\lfloor \frac{n}{2}+1\right\rfloor \right)\approx \left(\begin{array}{c} n-1\\ \left\lfloor \frac{n}{2}\right\rfloor \end{array}\right){\mathrm{FP}}_{1}{\mathrm{FP}}_{2}^{\left\lfloor \frac{n}{2}\right\rfloor },$$
(17)$${\mathrm{FN}}_{\mathrm{conf}}\left(n,\left\lfloor \frac{n}{2}+1\right\rfloor \right)\approx {\mathrm{FN}}_{1}+\left(1-{\mathrm{FN}}_{1}\right)\left(\begin{array}{c} n-1\\ \left\lfloor \frac{n+1}{2}\right\rfloor \end{array}\right){\mathrm{FN}}_{2}^{\left\lfloor \frac{n+1}{2}\right\rfloor }.$$

If *n* is odd, these expressions are equal to the ones provided in Section 3.4 of Results. Appendix A shows the adequacy of these approximations.

### 2.4. Simulation of Clinical Trial with Repeated Sampling for Self-Resolving Infections

For self-resolving infections, if a sample is taken time *t* after infection is first detectable, we assumed that the probability of detection follows a decaying exponential, $2^{-\left(t/t_{1/2}\right)}$, where $t_{1/2}$ is the infection detectability half-life. If infection is equally likely to occur at any time between sampling times (duration τ between samples over a period with uniform force of infection), the probability of detection at the first sample taken after infection is $\frac{1}{\tau }\int_{0}^{\tau }2^{-\left(t/t_{1/2}\right)}dt=\frac{2^{-\rho }\left(2^{\rho }-1\right)}{\rho \;\ln\;2}$, where ρ is the ratio $\frac{\tau }{t_{1/2}}$. If *N* samples are taken after infection (evenly spaced by τ), the probability that an infection is ever detected in at least one sample is
(18)$$P_{\mathrm{detection}}=1-\prod_{k=1}^{N}\left(1-\frac{2^{-k\rho }\left(2^{\rho }-1\right)}{\rho \;\ln\;2}\right).$$

To reflect the characteristics of a recent efficacy trial, 1400 participants (700 placebo arm, 700 vaccine arm) were simulated for a vaccine of 85% efficacy, annual incidence $p=0.02\times P_{\mathrm{detection}}$, false-positive rate FP = FP_1_ = FP_2_ = 0.0005, $t_{1/2}$ = 1.4 months, and the probability of detection is given by Equation (18). A range of inter-sample durations τ was tested, and incidence was converted to a per-sample basis, as described in Section 2.1. For convenience in the calculation of $P_{\mathrm{detection}}$, the value *N* was fixed to 20 (owing to the rapidly decaying infection, values in the range of 10 to 60 produce nearly identical results; if the infection is not detected in the first few assays after infection, it is likely to go undetected).

To calculate the expected trial duration, it was assumed that all participants were enrolled at the start of the trial, which proceeds until 24 infection cases are observed across both arms. This duration was estimated by dividing 24 by the expected overall case rate across both arms, which equals $700\times \left(p+\frac{\mathrm{FP}}{\tau }\right)+700\times \left(p\left(1-e\right)+\frac{\mathrm{FP}}{\tau }\right)$. If there is perfect case detection (*p* = 0.02) and assay specificity (FP = 0), expected trial duration is $24/\left(700\times 0.02+700\times 0.02\times \left(1-0.85\right)\right)=1.5$ years. If enrollments were, instead, staggered, the trial would be expected to continue until each participant had this follow-up duration on average.

### 2.5. Uncertainty in Parameter Values

A natural approach to model uncertainty in rate parameter *x* is to assume that an investigator started with an uninformative (uniform $\left[0,1\right]$) prior belief about that parameter and, then, performed *N* independent observations (i.e., *N* functionally independent samples, assuming a sufficiently reliable “gold standard”) to estimate that parameter. If a fraction $\bar{x}$ of these observations fall into the category of interest (e.g., the fraction observed to be false), the resulting posterior distribution for this parameter is $\mathrm{Beta}\left(1+N\bar{x},1+N\left(1-\bar{x}\right)\right)$ [9]. Higher *N* represents greater precision in the parameter estimates (a narrower beta distribution). Parameter values can then be drawn from this distribution, and all efficacy dilution calculations can proceed, as in the sections above.

For false-positive and false-negative rates, the value *N*, defining the uncertainty distribution, may be taken as the number of assay validation tests performed by the manufacturer using negative and positive controls, respectively. For the incidence, *N* may be treated as the total number of person-years observed in an epidemiological study. When relying on incomplete reports or complex epidemiological models to estimate incidence, the value *N*, used to define the distribution, may need to be chosen heuristically to reflect the number of “effectively independent” observations that the investigator believes has been made. If the relevant *N* for incidence is estimated from annual figures, the conversion from annual incidence to per-sample incidence $p=1-{\left(1-p_{annual}\right)}^{t}$ (with *t* the duration between samples, in years) occurs *after* drawing $p_{annual}$ from the beta distribution. The R script in Appendix A implements this approach.

## 3. Results

### 3.1. Overview

Vaccine efficacy is usually estimated by calculating the relative reduction in cases from the placebo to vaccine arms of a trial. Our results show that, in the presence of false positives that are not due to systematic error, the use of a “majority rule” replicate testing strategy can minimize the impact of case/non-case misclassification on estimates of vaccine efficacy.

For the purposes of this work, a “case” is assumed to be determined based solely on a diagnostic assay; this assumption ignores the role that symptom assessment can also play in defining cases. This assumption is particularly relevant for applications to infections that can have nonspecific symptoms, such as respiratory viruses, or infections that are asymptomatic. In these applications, the standard practice of keeping assay readers blinded to symptom data is important to eliminate potential bias.

Most assays are imperfect, so the number of truly infected subjects will generally differ from that determined by the assay, and similarly, it will differ for the number who are truly uninfected. We first show how assay false positives and false negatives can dilute the observed vaccine efficacy (i.e., reduce its apparent value, Section 3.2; Methods in Section 2.1) and, then, how this dilution can be counteracted by replicate testing strategies using “majority rules” (Section 3.3 and Section 3.4; Methods in Section 2.2 and Section 2.3). In particular, a strategy that uses replicates only to confirm an initial positive readout can efficiently counteract efficacy dilution. Next, by simulating a vaccine efficacy trial, we show how replicate testing can improve efficacy estimation and shorten clinical trial duration when investigating self-limiting infections that are detectable for a limited time (Section 3.5; Methods in Section 2.4). Finally, we offer an approach for selecting a replicate testing strategy when assay characteristics are not known with certainty (Section 3.6; Methods in Section 2.5).

### 3.2. Efficacy Dilution Is Sensitive to the False Positive Rate—And Less to the False Negative Rate

The dilution, or fractional reduction, in observed (estimated) vaccine efficacy, due to false positives and false negatives in a diagnostic assay, equals (Methods, Equation (8))
$$D=\frac{\mathrm{FP}}{\mathrm{OIP}}=\frac{\mathrm{FP}}{\mathrm{TIP}\left(1-\mathrm{FN}\right)+\left(1-\mathrm{TIP}\right)\mathrm{FP}}\approx \frac{\mathrm{FP}}{\mathrm{TIP}+\mathrm{FP}},$$
where FP and FN are the assay false-positive and false-negative rates, respectively, OIP is the observed incidence of infection in the placebo arm, and TIP is the true incidence of infection in the placebo arm. Methods Section 2.1 provides additional details about the definition of these terms. The approximation in this formula holds well if FP, FN, and TIP are no more than 0.2.

Figure 1 shows that vaccine efficacy is considerably underestimated if the false-positive rate is large compared to the incidence (in an unvaccinated population). A false-negative rate up to 0.2, however, has little effect regardless of incidence. This figure also shows the adequacy of the above approximation, which understates the exact value by no more than 12.5 percentage points, as long as FP, FN, and TIP are, themselves, no more than 0.2. Note that FN is not included in this approximation, which is consistent with the relative insensitivity of dilution to false-negative rate.

### 3.3. “Majority Rule” Replicate Assay Strategy Improves Estimation of Vaccine Efficacy

Replicate testing is one common strategy to improve assay characteristics. We can consider a compound testing strategy that uses multiple replicates and a rule to adjudicate the outcome as a single test. The performance characteristics of this single compound test is then referred to as “effective FP” or “effective FN”.

If an odd number *n* of independent replicates, of the same assay, are performed and the majority result is used (“majority rule”), the effective false-positive and false-negative rates are both reduced according to the approximate formula, which is usable for rates below 25% (Section 2.1, Equation (13)):$$\mathrm{Effective}\;\mathrm{false}\;\mathrm{positive}\;\mathrm{or}\;\mathrm{negative}\;\mathrm{rate}\;\mathrm{in}\;“\mathrm{majority}\;\mathrm{rule}”\;\mathrm{scheme}\approx \left(\begin{array}{c} n\\ \left(n+1\right)/2 \end{array}\right)x^{\frac{n+1}{2}},$$
where *x* is either the false-positive or false-negative rate; for odd *n*, the equation is identical for both.

Figure 2 illustrates that replicate testing strategies lower the dilution (improve estimation) of efficacy markedly: Increasing the total number of replicate assays reduces the false-positive rate, leading to recovery of the observed efficacy. Equations (9)–(12), in the Methods section, describe the more general situation where *n* may be an even number or where criteria other than majority rule are used.

### 3.4. “Confirmatory Majority Rule” Strategy Estimates Vaccine Efficacy Efficiently

Given that the dilution in observed efficacy is relatively insensitive to the false-negative rate, a more efficient strategy that decreases effective FP while increasing effective FN may be acceptable in some studies. In such situations, an initial assay can be used to identify provisional cases, with confirmatory assays performed only on replicate samples from these cases. A confirmed case is declared if a majority of all assays (including the initial one) are positive. For example, if a sample first tests negative, it is considered negative with no further investigation, but if it tests positive, it is tested twice more and labeled as positive if (and only if) there is a positive result on at least one of the two confirmatory tests (and, thus, on two of the three tests in total).

If FP_1_ and FN_1_ are the false-positive and false-negative rates for the initial assay and FP_2_ and FN_2_ are the rates for the confirmatory assays, the effective false-positive and false-negative rates for the testing strategy with *n* − 1 confirmatory assays (for odd n≥3) are approximately equal to (Section 2.1, Equations (16) and (17)):$$\mathrm{Effective}\;\mathrm{false}-\mathrm{positive}\;\mathrm{rate}\;\mathrm{in}\;“\mathrm{confirmatory}\;\mathrm{majority}\;\mathrm{rule}”\;\mathrm{strategy}\approx \left(\begin{array}{c} n-1\\ \left(n-1\right)/2 \end{array}\right){\mathrm{FP}}_{1}\;{\mathrm{FP}}_{2}^{\frac{n-1}{2}},$$
(19)$$\mathrm{Effective}\;\mathrm{false}-\mathrm{negative}\;\mathrm{rate}\;\mathrm{in}\;“\mathrm{confirmatory}\;\mathrm{majority}\;\mathrm{rule}”\;\mathrm{strategy}\approx {\mathrm{FN}}_{1}+\left(1-{\mathrm{FN}}_{1}\right)\left(\begin{array}{c} n-1\\ \left(n+1\right)/2 \end{array}\right){\mathrm{FN}}_{2}^{\frac{n+1}{2}}.$$

The confirmatory strategy, using the same assay for each replicate (FP_1_ = FP_2_, FN_1_ = FN_2_), can achieve nearly the same effective false-positive rate as the simple majority rule strategy does (Figure 2, blue versus red curves). As a single negative in the first assay always leads to a negative result in the confirmatory strategy, this strategy exhibits higher effective false negatives than even the single-replicate strategy. Nonetheless, as efficacy dilution primarily depends on false positives, the confirmatory strategy can prevent efficacy dilution almost as well as the simple majority rule does, while using fewer assays overall. If the elevated effective false-negative rate is undesirable, the use of a more sensitive initial assay in the confirmatory strategy (FP_1_ > FP_2_, FN_1_ < FN_2_) can control this rate at the cost of some efficacy dilution (Figure 2, green versus blue points).

### 3.5. Replicate Assay Strategy Can Improve Efficacy/Detection Trade-Off in Clinical Trials of Vaccines for Self-Limiting Infections

Many infections—such as respiratory viruses, herpesviruses (CMV, EBV), HPV, or Borrelia (Lyme disease)—are detectable for a limited time before the infection ends or enters latency. This limited detection period may necessitate frequent longitudinal sampling to count cases in vaccine trials. As frequent sampling also provides opportunities for false-positive results to accrue, an assay with high specificity can, nonetheless, dilute efficacy to unacceptable levels.

When longitudinal sampling is used, the results in previous sections are still valid, but they now apply to each sampling occasion. In particular, incidence needs to be defined per the duration between samples. For example, if an assay is used every month to detect an infection with placebo incidence of 2% per year, the corresponding incidence of 0.168% per month must be used, instead, to calculate efficacy dilution. If this assay has a false-positive rate of 0.05%, no false negatives, and if no cases are missed due to the decay of pathogen biomarkers, then a monthly testing regimen would dilute a true efficacy of 85% to an observed efficacy of 66% ($=85\%\times \left(1-\frac{0.05\%}{0.168\%+0.05\%}\right)$, see Section 2.1, Equation (8). Annual testing by the same assay would only dilute efficacy to 83% ($=85\%\times \left(1-\frac{0.05\%}{2\%+0.05\%}\right)$.

Pathogen biomarkers that decay over time therefore present a trade-off for clinical trial design: While less frequent sampling can reduce efficacy dilution, it also can miss cases, ultimately leading to a longer trial duration to collect the number of cases required for statistical analysis.

We simulated an efficacy trial to illustrate this trade-off, using parameters that reflect characteristics of a vaccine, infection, and assay, for a recent clinical development program (Figure 3). In this scenario, if only a single assay is performed on each sample, frequent sampling is vulnerable to substantial efficacy dilution due (as above) to the possibility of FP or FN in each sample and due to the aforementioned potential accumulation of those errors.

While reducing sampling frequency to less than monthly is likely to improve efficacy estimation, it does so, potentially, at the cost of a longer trial (assuming, as is common, that a set number of cases need to be accumulated). Since this infection only has a limited window of detectability, reduced sampling frequency also reduces the fraction of infections that are detected, effectively reducing observed incidence and contributing to efficacy dilution. Considering these effects together, observed efficacy in this simulated scenario can never exceed 74% as long as a single assay is used on each sample. The confirmatory majority rule strategy with 3 replicate assays can, however, resolve this trade-off by reducing the effective false-positive rate 1000-fold. This strategy enables the estimation of the efficacy without extending trial duration.

### 3.6. Managing Uncertainty in Incidence and Assay Characteristics

By drawing parameter values from uncertainty distributions around each input parameter, the above analyses can also be performed in typical scenarios where true incidence, false-positive, and false-negative rates are estimated, instead of known exactly (see Section 2.5). This approach generates an uncertainty distribution for the efficacy dilution, which may, itself, be used as an input for the calculation of study power and other analyses. The R script that implements this approach is provided in Appendix A.

Modest uncertainty can meaningfully impact study power: In the scenario envisioned in Figure 4 (FP 0.03, FN 0.2, TIP 0.02), when using a confirmatory majority rule strategy with 3 replicate assays, a true 80% vaccine efficacy is diluted to 72%, but accounting for parameter uncertainty, the diluted efficacy may be as high as 78% or as low as 51% (range of 95% CI in the bottom center panel). In the context of a vaccine development program, acknowledging this asymmetric downside risk may prompt a re-thinking of study design to avoid unwarranted no-go decisions.

## 4. Discussion

In any vaccine efficacy trial with a clinical endpoint, false-positive reports of infection, even when evenly distributed among active and control arms, tend to reduce (“dilute”) the estimated vaccine efficacy. While it is possible to adjust the estimate of efficacy to correct for this bias [7], doing so requires multiplying the result by a factor greater than one, which then also increases the variance of the estimate, thus requiring a larger sample size to maintain the same level of statistical power for a desired endpoint. While false positives and false negatives have been treated equally in the derivations, the results show that a given false-negative rate generally dilutes (biases) the efficacy estimate less than would be expected for a similar false-positive rate (Figure 1).

This article provides the first systematic analysis of how replicate assay measurements may be used, pragmatically, to prevent efficacy dilution and improve statistical power to distinguish between study arms. A “majority rules” replicate assay strategy can enable a clinical trial that would otherwise be doomed by unacceptable efficacy dilution. By requiring multiple positive results in order for an infection case to “count,” this strategy can reduce the false-positive rate by orders of magnitude and move efficacy estimates toward the true value (Figure 4, left column). If the cost of performing multiple assays at every sampling event is too high, then nearly the same outcomes can be achieved by performing replicate assays only where an initial assay is positive (Figure 4, middle and right columns). A confirmatory strategy uses resources efficiently and does not substantially increase the false-negative rate; it may be preferred to the implementation of an assay with higher intrinsic specificity.

Conditional (confirmatory) replicate testing requires additional assay replicates only for samples that are positive on their first test. This means that the total number of extra sample tests is expected to be a small fraction of the total number of samples (OIP, Equation (1)), which is to say a marginal cost. (If retesting a sample is not conditioned on the outcome of the first test so that every sample is tested multiple times, assay costs could increase by a factor of the number of replicates; the results presented suggest that this additional cost is unlikely to be justified by any noticeable benefit, if any, over the proposed conditional method.) Moreover, in all the replicate strategies, the additional replicates should use aliquots of the original samples from the subjects, so no additional clinic visits or sample collection would be needed; this would prevent any additional burden on the trial subjects. This aliquoting assumption might not hold for all types of sample matrices and assays or if the number of replicates (aliquots) is too large to be obtained from a single sample.

The challenge posed by efficacy dilution is magnified for trials in which frequent sampling is required to detect cases, as false-positive cases can also be introduced by frequent sampling. Vaccine development programs, for the prevention of infections that resolve rapidly in study participants, can encounter this challenge. The replicate assay strategies developed here can resolve this challenge by eliminating the trade-off between accurate efficacy estimation and the time required to detect needed cases (Figure 3).

The assay false-positive rate that could cause a “false no-go” decision can depend on many aspects of a vaccine development program, including the specific (efficacy-based) decision criteria, the variability in the population’s vaccine response, durability of any infection-related protection (in placebo arm) or boost effect (in active arms), incidence rates and force of infection, as well as other factors. Hence, this manuscript has focused on the general principles and calculations needed to minimize and characterize any risk associated with imperfect sensitivity and specificity of the diagnostic assay.

These general principles apply beyond vaccines and their efficacy. This analysis is also relevant to sampling for safety signals or other binary endpoints used to support decisions and which use a potentially fallible assay. For example, similar principles of frequent longitudinal sampling have also been applied to public health screening. Serial sampling has been recommended for rapid antigenic testing for SARS-CoV-2 to provide asymptomatic or presymptomatic individuals with timely information to prompt isolation [10]. In settings where community infection prevalence is low and the cost of isolation is high, majority rule testing may be an appropriate strategy to avoid unnecessary isolation on the part of frequent serial testers [11]. Replicate testing protocols have also been used to resolve concerning or discordant test results for at-home or point-of-care diagnostics, such as for prothrombin time testing [12]. When the consequence of a positive test may be life-altering—such as for HIV or cancer screening—public health agencies and physician organizations offer detailed practitioner guidance around such follow-up testing [13,14].

Ignoring uncertainty in the false-positive rate, false-negative rate, or true infection incidence can provide undue confidence in the ability of a clinical trial to estimate vaccine efficacy (Section 3.6). Uncertainty poses an asymmetric risk: while obtaining more false positives than expected can substantially reduce observed efficacy, obtaining fewer false positives than expected does not increase observed efficacy above the true value. This principle applies to public health studies as well: in a seroprevalence study of SARS-CoV-2, uncritical reliance on a point estimate of the assay false-positive rate may have led to the overestimation of population prevalence [15]. Effective planning for vaccine trials, therefore, requires complete and transparent accounting for uncertainty in endpoint assays. While our exposition has focused only on control arms that use a placebo, the same analysis applies to comparisons between two active arms, except that efficacy is replaced with the relative efficacy of the more potent vaccine. False positives can, therefore, bias trials against finding *either* superiority or inferiority of the test agent versus comparator.

The utility of replicate strategies depends, crucially, on the independence of false-positive outcomes. In other words, the fact that a single false positive has occurred when the assay is used for one sample is assumed not to make it more likely that a false positive will occur for another sample taken from the same person at the same time. Protocols for shuffling samples so that multiple samples from the same person are not likely to be assayed on the same plate or in the same batch may provide assurance of independence. Biological mechanisms that affect all samples from a study participant, e.g., nonspecific PCR amplification or presence of cross-reactive antibodies, would cause false positives that replicate testing cannot address.

The R Shiny app provided in Appendix A (and at https://github.com/Merck/FPFN) may assist investigators with the design of clinical trials using replicate assay strategies. Given an estimate of and degree of certainty in the false-positive, false-negative, and incidence rates, the app calculates the resulting posterior distribution of efficacy dilution and generates outputs similar to Figure 4. With this distribution in hand, the investigator may simulate clinical trials for a range of sampling strategies and select one that provides sufficient study power.

## Figures and Tables

**Figure 1 vaccines-11-01501-f001:**
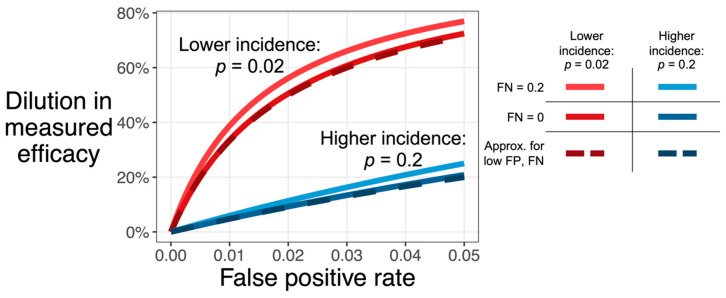
**Dilution in observed efficacy (D, y-axis) increases with the assay false positive rate (FP, x-axis).** The effect, shown here for a testing strategy that only tests once, is more extreme for infections with low incidence (red: TIP = 0.02) than for infections with high incidence (blue: TIP = 0.2). Dilution also increases with the false-negative rate (lighter color, FN of 0.2 versus darker color, FN of 0), though this effect is smaller than the effect of false positives. The dashed line shows the approximation $D\approx \frac{\mathrm{FP}}{\mathrm{TIP}+\mathrm{FP}}$, which assumes that both FP and FN are low (i.e., no greater than 0.2; see Section 2.1, Equation (8)).

**Figure 2 vaccines-11-01501-f002:**
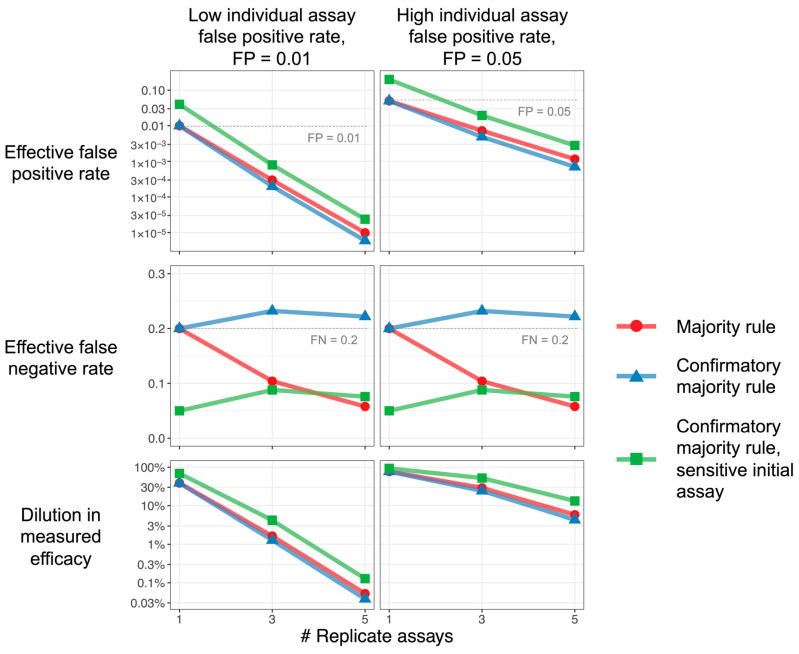
**Replicate assays improve efficacy estimation.** Characteristics of majority rule and confirmatory majority rule strategies, as functions of the number of replicate assays, are used. The majority and confirmatory majority rules, plotted in red and blue, use the same assay for each replicate. The confirmatory majority rule, plotted in green, uses a more sensitive (smaller FN) and less specific (higher FP) initial assay; subsequent replicates are all the same. Increasing the number of replicates for all strategies reduces the false-positive rate (top row), but the confirmatory strategy exhibits a higher false-negative rate unless a more sensitive initial assay is used (middle row). Since the dilution effect depends, primarily, on false positives, all strategies can achieve reductions in efficacy dilution (bottom row). *Parameters:* Incidence TIP = 0.02. Majority rule and confirmatory majority rule: FP = 0.01 (left column), 0.05 (right column); FN = 0.2. For confirmatory majority rule with sensitive initial assay, the initial assay has 4-fold reduction in false-negative rate and 4-fold increase in false-positive rate: FP for the first assay = 0.04 (left column), 0.2 (right column); FN for the first assay = 0.05. Note that, for a single replicate (1 on the x-axis), the value plotted is just that of a single assay, and the difference between the “sensitive initial value” result and others is only due to the different characteristics assumed for that assay.

**Figure 3 vaccines-11-01501-f003:**
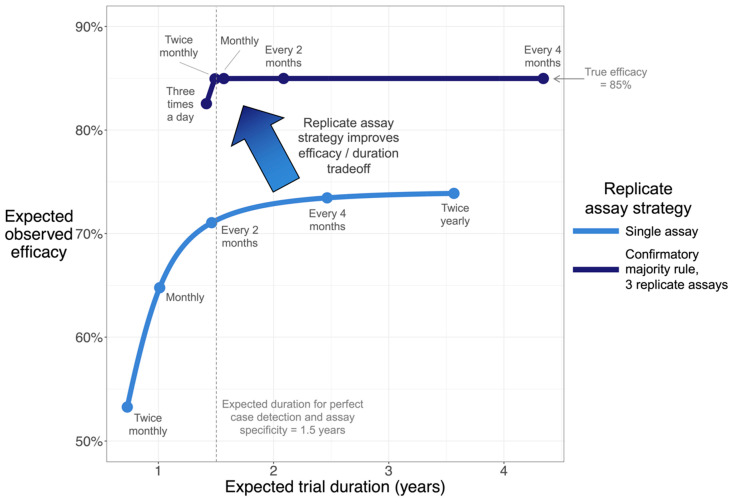
**Simulation shows that confirmatory majority rule accelerates clinical trials and improves efficacy estimation.** Curves plot trade-off between estimated efficacy and clinical trial duration, where longitudinal samples are used to detect infection. A vaccine efficacy trial for a self-limiting infection (resolved in a median of 1.4 months) was simulated, assuming true efficacy of 85%. The trial proceeds until a pre-specified number of cases are collected. Each curve was calculated by simulating a range of sampling frequencies. Dense simulations were performed to generate curves; points and accompanying text highlight selected sampling frequencies. In this scenario, the confirmatory majority rule strategy (upper curve) is only vulnerable to noticeable efficacy dilution at extreme sampling frequencies (e.g., three times a day). See Methods Section 2.4 for all parameters.

**Figure 4 vaccines-11-01501-f004:**
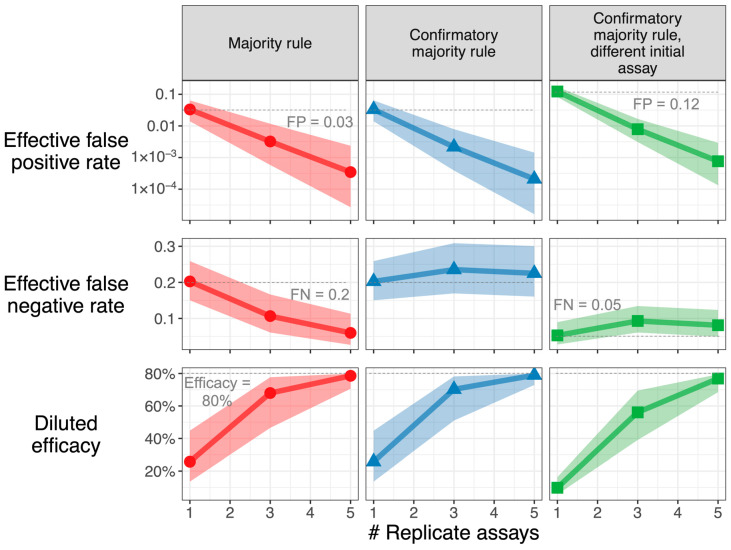
**Downside risk of parameter uncertainty is large.** Characteristics of sampling strategies, accounting for uncertainty in incidence, false positive, and false-negative rates, are used. The resulting median and 95% credible intervals are shown for effective FP, effective FN, and diluted efficacy, assuming a true vaccine efficacy of 80%. When the median estimate for diluted efficacy is near this true value, the credible interval is asymmetric with a larger downside, as the upper bound can never exceed the true value. *Parameters:* Incidence TIP has n expected value of 0.02. Majority rule and confirmatory majority rule: FP has an expected value of 0.03; FN has an expected value of 0.2. For confirmatory majority rule with sensitive initial assay, the initial assay has an (expected) 4-fold increase in false-positive rate (to 0.12) and a 4-fold reduction in false-negative rate (to 0.05) versus confirmatory assays (expected FP = 0.03, expected FN = 0.2). Parameter uncertainty is characterized by the beta distributions described in the Methods section, assuming that *n* = 1000, 200, and 200 observations were performed to estimate incidence, false-positive, and false-negative rates, respectively. The middle 95% of these beta distributions: TIP [0.013, 0.031], FP [0.014, 0.064], FN [0.151, 0.261], FP of sensitive initial assay [0.082, 0.172], FN of sensitive initial assay [0.028, 0.090]. For each sampling strategy, 10,000 sets of values were drawn from the uncertainty distribution. The R Shiny app in Appendix A can be used to recreate this graph for user-inputted parameters.

## Data Availability

All data for figures can be reproduced with the code provided in the Appendix A.

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
