# Peer review of "Replicate Testing of Clinical Endpoints Can Prevent No-Go Decisions for Beneficial Vaccines"

_vaccines, 2023, doi:10.3390/vaccines11091501_

Round 1

Reviewer 1 Report

Title: Replicate testing of clinical endpoints can prevent no-go decisions for beneficial vaccines

Summary: This manuscript is an analytic exploration of strategies, specifically replicate testing, to reduce the false positive rate in clinical trials of vaccines with the goal of reducing the dilution of results leading to no-go decisions in vaccine development.

Introduction:

·       The authors discuss how misclassification can lead to “unwarranted” no-go decisions which I would describe more as cautious decision making, and while what they state about the direction of bias is true they should discuss more what leads to no-go decisions as in many vaccine trials there is nuance to measurement (timing), prevalence, and decisions making

·       Can the author briefly discuss what level of misclassification is needed to drop vaccine efficacy rate below the no-go decision level, and whether in general diagnostic assays are accurate enough in those instances? Mainly want to know if there are there real world examples of this in the literature? How often would these methods be needed?

Results:

·       It appears the manuscript document was drafted in another program such as R and knitted to Word… there are a number of errors and missing values (see bullets below) throughout the manuscript making it challenging to read – Please read the word document carefully and fix all of these.

o   It would be helpful to reread this manuscript after all these missing values and error notices are resolved

·       Line 81, 83, 89, elsewhere state - (Sections 0; Methods in Sections 0) making it hard to follow what these sections are referencing

·       Lines 142, 216, elsewhere have error codes

·       Equations lines 141, 161, 164, elsewhere are missing a horizontal division line

·       Should equations in the results section also be numbered to differentiate them from the methods equations or help refer to the same equations referenced in methods

Materials and Methods

·       The materials and methods should go before the results – I initially read the results first very confused without definitions and equations of each strategy

·       Section 3.2 you use the term replicate assays but in results begin to use the term majority rule, I would include the term here so it’s clear that is what you are referencing

Discussion

·       Reference of Figure 3 lines 399 and 401 appear to be referencing the wrong figure, same for line 410

·       Paragraph 4 are there other examples here?

·       A further discussion of cost and burden to trial patients should be evaluated in the discussion

Reviewer 2 Report

Replicate testing of clinical endpoints can prevent no-go decisions for beneficial vaccines

1. Pag 4 lin 142 and Pag 6 lin 216 “ERROR. Reference source not found”.  

Correct the sentence that appears in bold

2. An explanation of the result appears in the title in Figures 1-4. The title of the Figures must be descriptive and the explanation must appear in the results of the publication.

Reviewer 3 Report

Authors raises concerns about the reduction of apparent efficacy (the dilution) of vaccines, which can lead to unwarranted no-go decisions in the vaccine development process or to misleading interpretations of vaccine effectiveness. To address this problem, authors suggest a series of “majority rule” replicate assay strategies that can counteract such dilution. Here, multiple positive results are required for an infection case to count. The strategies and the software provided here can be helpful in minimizing the chance of unwarranted no-go decisions for clinically beneficial vaccines, especially for many asyptomatic cases.

Despite the sophisticated approach and the value in saving resources, the readers would be helped a lot by having a summary of different scenarios (either in figures or tables). That summary may include specificity issues resulting from cross-reactivity. Also, the investigator biases toward the false-positives will have to be balanced with real-world data, if available.

Minor comment.

line 216 (Error! Reference source not found.). needs more explanation.

Round 2

Reviewer 1 Report

Introduction – Comments addressed

Methods – section order and errors have been addressed

Results – Line 220 - the discussion should include an description of how symptomatic cases would impact a case definition/false positive rate even if only nonspecific symptoms, as any presence of symptoms may bias case finding – unless blinded to symptoms this work would truly apply to asymptomatic infection Line 227 has an error message

Discussion – See note in results; all other comments have been addressed

Author Response

Both changes have been made. We are grateful for your thoughtful review of our manuscript.